# miRNA-146a and miRNA-126 as Potential Biomarkers in Patients with Coronary Artery Disease and Generalized Periodontitis

**DOI:** 10.3390/ma14164692

**Published:** 2021-08-20

**Authors:** Jaideep Mahendra, Little Mahendra, Hytham N. Fageeh, Hammam Ibrahim Fageeh, Wael Ibraheem, Hesham H. Abdulkarim, Anilkumar Kanakamedala, Prashanthi Prakash, Sruthi Srinivasan, Thodur Madapusi Balaji, Saranya Varadarajan, Raghunathan Jagannathan, Shankargouda Patil

**Affiliations:** 1Department of Periodontology, Meenakshi Ammal Dental College and Hospital, Chennai 600095, India; dranilkumar7979@yahoo.com (A.K.); prashanthisankar@gmail.com (P.P.); drsruthisvasan@gmail.com (S.S.); 2Department of Periodontics, Maktoum Bin Hamdan Dental College, Dubai 213620, United Arab Emirates; littlemahendra24@gmail.com; 3Department of Preventive Dental Sciences, College of Dentistry, Jazan University, Jazan 45142, Saudi Arabia; hfageeh@jazanu.edu.sa (H.N.F.); hafageeh@jazanu.edu.sa (H.I.F.); Wibraheem@jazanu.edu.sa (W.I.); 4Advanced Periodontal and Dental Implant Care, Missouri School of Dentistry and Oral Health, A. T. Still University, St. Louis, MO 63104, USA; heshamabdulkarim@atsu.edu; 5Department of Periodontology, Tagore Dental College and Hospital, Chennai 600127, India; tmbala81@gmail.com (T.M.B.); doctorraghunathan@gmail.com (R.J.); 6Department of Oral Pathology and Microbiology, Sri Venkateswara Dental College and Hospital, Chennai 600130, India; vsaranya87@gmail.com; 7Department of Maxillofacial Surgery and Diagnostic Sciences, Division of Oral Pathology, College of Dentistry, Jazan University, Jazan 45142, Saudi Arabia

**Keywords:** biomarkers, periodontitis, coronary artery disease, micro-RNA, subgingival plaque

## Abstract

The present study aims to compare the levels of micro-RNA-146a and micro-RNA-126 in oral subgingival plaque and coronary plaque from artery walls in patients with coronary artery disease who suffer from generalized periodontitis. A total of 75 participants were selected and grouped into three categories of 25 patients each: GP+CAD, GP, and HP groups. GP+CAD consisted of patients diagnosed with generalized periodontitis (GP) and coronary artery disease (CAD). The GP+CAD group was further divided into two groups—GP+CADa: where subgingival plaque samples were collected; GP+CADb group: where coronary plaque samples were collected while the patient underwent a coronary artery bypass grafting surgery. The GP group consisted of 25 patients diagnosed with only generalized periodontitis. The HP group consisted of 25 systemically and periodontally healthy controls. miRNA-146a and miRNA126 levels were assessed in subgingival plaque (SP) samples from all groups. Results revealed that miRNA-146a was expressed at higher levels and miRNA-126 was downregulated in the GP+CAD group. microRNAs in subgingival plaque samples showed a significant correlation with the coronary plaque samples in the GP+CAD group. miRNA-146a and miRNA-126 were present in coronary artery disease patients with periodontitis. These micro-RNAs may serve as risk biomarkers for coronary artery disease and generalized periodontitis.

## 1. Introduction

Micro-RNAs are a family of small non-coding RNA molecules that regulate gene expression [1]. They are found in tissues, plasma, and other body fluids in a stable form that is protected from endogenous RNase activity. In the innate immune system, micro RNAs are the first line of defense against microorganisms [2]. More than 1000 human miRNAs have been identified. Protein expressions are regulated by miRNAs. They bind to complementary sites located on target messenger RNA molecules and inhibit the translation of mRNA into proteins, repressing the target genes [3]. MiRNAs play a vital role in periodontal cellular functions through degradation and translational repression of mRNAs. In tissues, miRNAs regulate the gene expression involved in differentiation, growth, proliferation, and apoptosis. Various chronic ailments, including periodontitis and cardiovascular disease, can occur due to the alteration in miRNA signatures. miRNA may have applications in diagnosis and therapeutic strategies in treating patients suffering from these inflammatory disorders [4].

Within the periodontium, the mRNAs play a significant role against bacterial insult. They initiate the immune response via Toll-like receptors and release of proinflammatory cytokines such as IL-1, IL-6, and TNF-α. These transcriptomes are upregulated in the inflamed periodontium and regulate the innate and adaptive immune response in periodontal inflammation [5].

Previous research has linked miRNAs to a regulatory role in the pathogenesis of periodontitis. In human gingival fibroblasts, miRNA-146 plays a crucial role in inhibiting pro-inflammatory cytokines through IL-1 receptor-associated kinase-1 (IRAK1) [6]. It also plays a significant role in cardiovascular diseases, relating to inflammatory cascades and mechanisms of LDL release from the liver. Studies show that miR-146a levels are elevated in the aortic and femoral plaques of cardiac patients [7]. Research has linked mRNA146a in the pathobiology of periodontal disease and coronary heart disease. So far, no study has sought to explore its role as a specific and sensitive biomarker associating chronic periodontitis with coronary heart disease. 

MicroRNA-126 (miR-126) is located in an intron of the epidermal growth factor-like-domain 7 gene (*EGFL7*) and is specifically and highly expressed in the endothelial cells (ECs), which regulates ECs migration, cytoskeleton reorganization, capillary network stability, cell survival, and apoptosis [8]. MicroRNA-126 regulates cell survival or apoptosis in various cell types. MiR-126 is crucial for the maintenance of the vascular structure in vivo [9]. Studies have shown that mRNA-126 can evaluate the risk for CAD and early atherosclerosis [8,9]. It remains to be determined whether the down-regulation of miR-126 in CAD patients is directly involved in inflammation or if it is a compensatory response to this process. Owing to the significant role of miR-126 level in association with LDL cholesterol, the circulating miRNA levels may reveal a compensatory response to inflammation.

Periodontitis and cardiac disease have shown the upregulation of miRNA-146a. miRNA-126 has been shown to downregulate the proinflammatory cytokines such as tumor necrosis factor-α (TNF-α) and interleukin-β (IL-β). Within the cardiac tissues, miRNA-126 maintains the integrity of the blood vessels and protects against ischemia-induced blood vessel deterioration [10]. It plays a dual role in the atherosclerotic pathway downregulating VCAM-1 expression; thus, limiting the adherence of leukocytes and plaque formation. However, it also upregulates smooth muscle cell proliferation and activation, which may exaggerate the atherosclerotic process [10].

Research on genetic associations between these conditions is still in its nascent stages. To date, few studies have examined the association of these transcriptomes in subgingival and coronary plaque samples of generalized periodontitis patients associated with CAD. The importance of miRNA-126 in CAD and periodontal disease has not been closely examined. miRNA126 may affect the upstream signaling of NF-κB which impairs cytokine production. miRNA-146a and miRNA-126 have been analyzed extensively in the various inflammatory conditions based on their specificity and sensitivity. Yet, their potential role in periodontal inflammation in CAD remains understudied. This study is the first to analyze miRNA-146a and miRNA-126 from inflamed periodontal tissues of CAD patients. This paper explores the relation between demographic factors and periodontal clinical parameters such as BMI, age, plaque index, probing pocket depth, clinical attachment loss, along with the lipid profile, including Total Cholesterol, High-Density Lipoprotein-c, Triglyceride, LDL-c, systolic, and diastolic blood pressure in periodontitis patients with CAD. The primary aim of this paper was to quantify micro-RNA-146a and micro- RNA-126 in dental plaque of generalized periodontitis and coronary plaque from CAD patients and to compare them against periodontitis patients without coronary artery disease. 

## 2. Materials and Methods

The present investigation was conducted between August 2017 and June 2018 in Chennai, Tamil Nadu, India. A total of 105 participants were included. A total of 75 participants were selected and grouped into 3 categories based on inclusion and exclusion criteria: GP+CAD, GP, and HP groups (Figure 1). GP+CAD group consisted of 25 patients diagnosed with generalized periodontitis and coronary artery disease (CAD). This group was further subdivided into GP+CADa and GP+CADb based on the nature of the plaque collected. GP+CADa group—where subgingival plaque samples were collected; GP+CADb group—where coronary plaque samples were collected while the patient underwent coronary artery bypass grafting surgery (CABG). GP group consisted of 25 patients who were systemically healthy but suffered from generalized periodontitis. HP group consisted of 25 systemically and periodontally healthy controls.

Inclusion criteria included: (1) Patients volunteering for the investigation. (2) Patients between the age group of 30 and 70 years. (3) Presence of ≥15 natural teeth. (4) GP and GP+CAD group included subjects with 30% or more sites having clinical attachment loss (CAL) ≥ 2 mm, radiographic evidence of alveolar crestal bone loss ≥ 2 mm from the cementoenamel junction. The GP+CAD group had coronary artery disease. (5) HP group included systemically and periodontally healthy subjects.

Exclusion criteria: (1) Patients associated with systemic diseases such as lung disorders, renal disease, liver disease, type I, type II diabetes mellitus, HIV infection, rheumatoid arthritis, allergy, or any malignancy. (2) Patients undergoing treatment with antibiotics, immunosuppressive drugs such as corticosteroids, within 6 months. (3) Female patients were excluded due to hormonal changes which might affect the oral flora. (4) Smoking and other tobacco usage habits. (5) Patients who underwent periodontal treatment within 6 months.

The present study was approved by the “Institutional Review Board of Meenakshi Ammal Dental College and Hospital (MADC/IRB-XXVI/2018/415)”. The protocol of the research was registered with the Clinical Trials Registry (ID No: NCT04583085). Written informed consent was obtained from all subjects.

The power of the study was obtained as 90% with 75 participants for sample size considering the previous literature and prevalence of both cardiac and periodontal disease in the location.

Demographic data acquired from the patient’s medical record included: age (30–70 years), BMI (Body Mass Index), and socioeconomic status based on monthly income.

In all three groups, two individually trained investigators were involved in assessing the following parameters: (1) Plaque Index (PI) [5], (2) bleeding on probing (BOP) [11], (3) probing pocket depth (PPD), and (4) Clinical Attachment Loss (CAL). Intra-examiner reliability was estimated as 0.73. A William’s probe (Williams probe, Hu-Friedy, Chicago, IL, USA) was used to record the periodontal parameters. 

Criteria for periodontitis: Diagnosis of periodontitis was confirmed when the patients had at least 30% or more sites with clinical attachment loss (CAL) ≥ 5 mm, at least four sites per tooth were present in the entire dentition with positive BOP [11], PPD of ≥4 mm [12]. For systemically healthy patients, (HP group), PPD ≤ 3 mm in every site with no BOP was taken into account. 

Systolic (SP) and Diastolic blood pressure (DP), High-density lipoprotein levels (HDL-c), Low-density lipoprotein levels (LDL-c), Total triglyceride levels (TG), and Total cholesterol levels (TC) were obtained in all three groups. 

The parameters were recorded at the time of admission for CABG for the GP+CAD group, before the periodontal treatment protocol for the GP group, and during the master health check-up for the HP group. Coronary plaque samples (GP+CADb subgroup) were obtained by the cardiothoracic surgeon at the time of coronary artery bypass grafting surgery (CABG). Subgingival plaque samples were obtained from the deepest periodontal sites using Gracey’s curette. 

The obtained plaque samples were then homogenized and stored in RNA LATER solution n at −80 °C till molecular analysis. MicroRNA146a and microRNA126 were isolated from coronary plaque and subgingival plaque samples using miRNA isolation kit (Thermo Fisher Scientific, Waltham, MA, USA). Nanodrop was used to quantify microRNA yield. Absorbance (A) was measured at 260/280 nm and 230/280 nm. Using the TaqMan RNA Reverse transcription kit (Applied biosystems, Thermo Fisher scientific, Waltham, MA, USA), MicroRNA underwent reverse transcription. Real-time PCR (CFX96 touch real-time PCR detection system, Bio-Rad, Hercules, CA, USA) was used to determine the values of miRNA146a and miRNA-126 in all three groups. In the control group, the expression was considered as a consistent value of 1 which was in concurrence with Schmittgen et al. [13]. The final results showed the expression of a fold increase in miRNAs. 

## 3. Statistical Analysis

Statistical estimation was conducted using SPSS version 25.0 (SPSS). Median and interquartile values were obtained for all the parameters in three groups along with the level of significance. A Kruskal–Wallis test was performed to obtain the difference in the variables between the three groups. A post hoc analysis was performed for multiple comparisons of both the biomarkers. Pearson’s correlation was performed to correlate the expression of microRNA-146a and miRNA-126 in the dental and coronary plaque of patients with GP+CAD. The level of significance was at *p* < 0.05.

## 4. Results

Demographic variables and clinical parameters of the periodontium, lipid profile, height, and BMI showed significant differences when compared among the three groups. Age and socioeconomic status were not significant. Parameters of the periodontium such as the Plaque Index (PI), probing pocket depth (PPD), bleeding on probing (BOP), and clinical attachment loss (CAL) also showed statistical significance among all the three groups. Cardiac parameters—TC HDL, TG, LDL—were insignificant. Random blood sugar (RBS) and blood pressure (BP) were significant among the groups (Table 1).

The levels of microRNA146a and microRNA126 in the subgingival plaque samples between the three groups were significant (Table 2a and Table 3a). For both the microRNAs, the post hoc analysis showed a significant difference between the (GP+CAD)a and HP group as well as between the GP group and HP group (*p* value < 0.001) (Table 2b and Table 3b). There was no statistically significant correlation between microRNA-146a (r-value −2.40, *p*-value 0.24) and microRNA-126 levels of subgingival and coronary plaque samples between (GP+CAD)a and (GP+CAD)b. r-value−0.12, *p*-value −0.57) (Table 4).

## 5. Discussion

Two decades of research have revealed an association of periodontitis between coronary heart disease. In this paper, we sought to shed light on this association at the molecular level. We quantified microRNA-146a and microRNA-126 levels in the coronary and subgingival plaques of the patients undergoing coronary artery bypass surgery procedures and compared them with the periodontitis patients without coronary artery disease. The two micro RNAs were correlated with each other in both groups. In all the groups, the median age was found to be insignificant since the participants were in the same age group (50 to 60 years) (Table 1). People over the age of 35 showed a higher risk and were more likely to suffer from periodontal and cardiovascular disease. Gjermo et al. demonstrated that the prevalence of moderate and severe periodontal disease was higher with the increase in age [14]. BMI in the GP+CAD, GP, and HP groups was found to be 24, 24.52, and 21.35 kg/m^2^, respectively, which showed a significant difference (Table 1). Studies by Lamon-Fava S et al. [15] and Flint A et al. [16] demonstrated a link between cardiac disorders and obesity. They found that patients with a higher BMI and coronary artery disease tended to have higher morbidity and mortality rates.

The median incomes in groups GP+CAD, GP, and HP were Rs. 30,000, 26,000, and 30,000, respectively. However, there was no statistical difference between the groups (Table 1). A study by Schultz M et al. [17] reported that a lower socioeconomic status was associated with a higher risk of cardiac disorders. De Mestral C et al. [18] echoed these findings claiming that in higher-income countries, socioeconomic status is a determinant of cardiovascular risk. In our study, the participants were mainly from an urban population that belonged to middle and developed socioeconomic backgrounds. This may explain the lack of statistical significance. 

Parameters of the periodontium such as the plaque index (PI), probing pocket depth (PPD), bleeding on probing (BOP), and clinical attachment loss (CAL) were found to be similar in the two groups GP+CAD and GP. They were statistically significant when compared to HP. These findings correlated with earlier results from Mattila et al. [19] and Bokhari et al. [20] who found increased periodontal destruction in patients with coronary artery disease and generalized periodontitis as compared to healthy patients. This increased periodontal destruction may be attributed to inflammatory markers such as acute phase reactants (cardiovascular disease markers). They may influence local periodontal inflammation leading to systemic illnesses. Microbial, inflammatory, and cellular mechanisms of periodontal disease can raise the risk for systemic diseases. Parameters of the periodontium were found to be similar in the GP+CAD and GP groups. This reflected the underlying biological mechanisms that correlate the inflammatory nature of both diseases (Table 1).

TG, TC, HDL, and LDL were similar among the groups and not statistically significant. Tang et al. [21] emphasized that an elevated cholesterol level is linked to the initiation and progression of coronary diseases. It may play a role in the mechanisms affecting the periodontal condition of a person (Table 1).

Blood pressure was found to be significant among the groups. The GP+CAD group had higher blood pressure as compared to the other groups. Desvarieux et al. [22] found an association between increased blood pressure, pathogenic oral microbiota, and periodontal parameters. In the present study, we observed that the subjects with cardiovascular disease had an elevated blood pressure. This may be due to factors such as a higher BMI and lifestyle.

miRNA-146a levels in GP+CAD showed a higher mean value (2.55 ± 0.74) as compared to GP and HP (2.24 ± 0.54 and 1.21 ± 0.25). GP+CADa and GP groups also expressed a significant difference in microRNA146a levels when compared to HP (Table 2a,b). Raitoharju et al. and Yagnik K et al. [23,24] demonstrated that microRNA-146a expression was elevated in the coronary plaque of CAD patients. This was due to its action on interleukin-1 receptor-associated kinase 1, TNF receptor-associated factor 6, and TLR-4 molecules. Motedayyen et al. [25] found that miRNA-146a expression was increased in tissues of generalized periodontitis patients when compared to the HP control group. They postulated that increased microRNA-146a caused the downregulation of the epidermal growth factor receptor. This led to disturbance in the transforming growth factor-β signaling; thus, retarding the regeneration of gingival tissues.

miRNA-126 were decreased in GP+CADa and GP when compared to HP. On intergroup comparison, GP+CADa and GP showed a significant difference compared to HP (*p* < 0.001). Our results are in agreement with previous research by Wu et al. who found that the overexpression of miRNA126 decreased the inflammatory response of human gingival fibroblasts through a reduction in the secretion of chemokine C-C motif ligand 2 (CCL2), TNF-α and IL-6 [26]. Wang et al. observed that miRNA-126 was down-regulated in patients with coronary artery disease [27]. They found that the expression of miRNA126 in blood circulation was related to the severity of coronary artery disease. Our results were consistent with those of Li who found an increase in the IL-6 and TNFα and a decrease in the miRNA126 expression in endothelial progenitor cells [28]. Mi RNA-126 may downregulate the elevated plasma levels of placenta growth factor (PLGF) in cardiac patients. GP+CAD showed an increased level of miRNA-146a with the downregulation of miRNA-126 in our study (Table 3a,b). This may be due to the inflammatory nature of coronary artery disease.

The correlation of microRNA-146a and microRNA-126 between groups GP+CADa and GP+CADb was found to be statistically insignificant (*p* value 0.24 and 0.57, respectively) (Table 4). According to Wu and Yang, miRNA-126 was downregulated in CAD patients. This suggests that it affects endothelial cell activities and inflammatory responses, which are pathological features of CAD and periodontitis [29]. We found a higher expression of microRNA-146a and the downregulation of mi RNA-126 among the groups. There was no apparent correlation between the HP and GP plaque samples with the GP+CAD group. Prior studies have established miRNA126 as one of the most endothelially enriched and anti-inflammatory biomarkers. It suppresses the human gingival fibroblasts by targeting cytokines. Previous studies have found a decrease in miRNA-126 with the elevation in ROS and TNF-α. miRNA-126 downregulates the pro-inflammatory cytokines. This could form the basis for the use of mi RNA-126 in target therapy. Further research should be undertaken to explore the interplay of these micro RNAs.

One limitation of our study was that no periodontal intervention was carried out which could have reflected on the existence of the biomarkers before and after periodontal treatment. Another limitation was that our study followed a cross-sectional design. Future investigations that are multi-centered and longitudinal studies with periodontal interventions could establish these transcriptomes as distinct, definitive, and reproducible biomarkers in the pathogenesis of periodontal disease and coronary artery disease.

## 6. Conclusions

The main goal of our study was to compare the levels of micro-RNA-146a and micro- RNA-126 in oral subgingival plaque and coronary plaque in patients suffering from coronary artery disease and generalized periodontitis with healthy patients. We found the upregulation of miRNA-146a and the downregulation of micro-RNA-126 in patients with coronary artery disease and periodontitis as compared to healthy controls. This finding suggests that miRNA-146a and miRNA-126 are present and involved in the disease process of periodontitis and coronary artery disease. This research provides insights into how these transcriptomes may serve as the potential risk biomarkers for both generalized periodontitis and cardiovascular diseases. The present study will serve as a base for future studies into therapy targeting the host immuno-inflammatory response.

## Figures and Tables

**Figure 1 materials-14-04692-f001:**
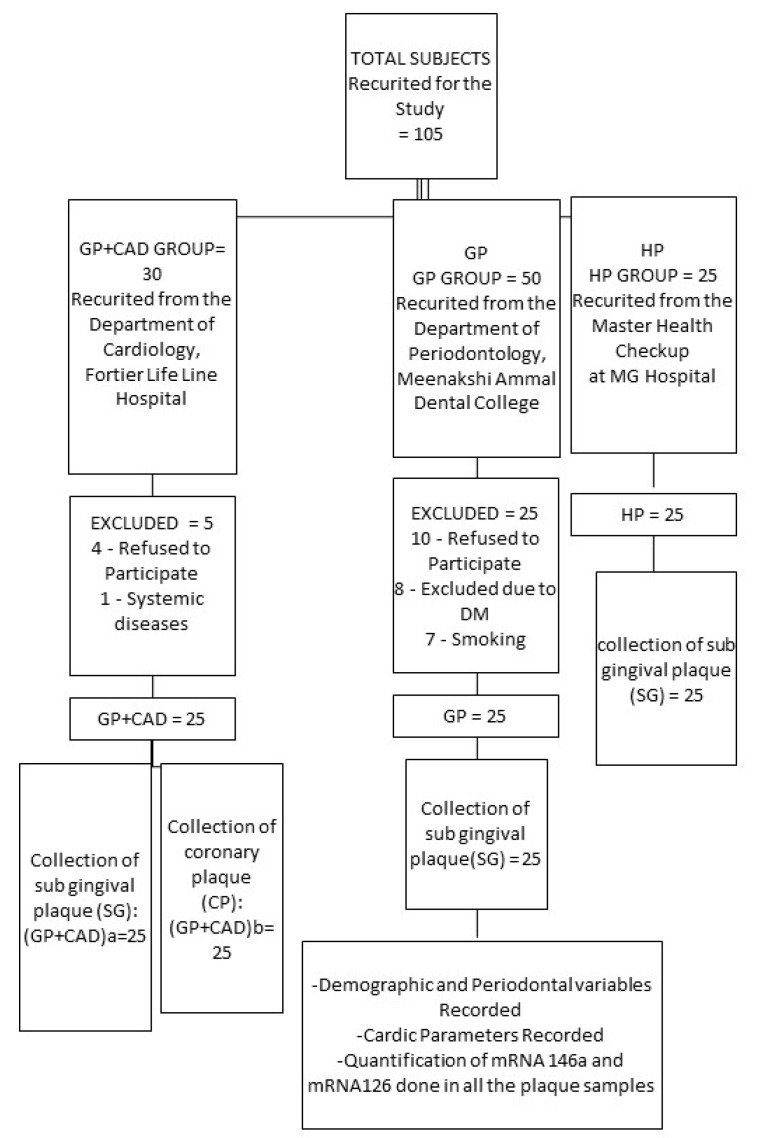
Flowchart 1—study design.

**Table 1 materials-14-04692-t001:** Comparison of the demographic variables, clinical parameters, and cardiac parameters among the three groups.

Variables	Median (I Quartile, III Quartile)			F	P
	GP+CAD	GP	HP		
Age(years)	59.00	51.00	50.00	5.18	0.07 ^NS^
Height	166.00	164.00	163.00	0.05	0.97 ^NS^
Weight	66.00	64.00	58.00	8.36	0.01
BMI (kg/m^2^)	24.00	24.52	21.36	8.55	0.01 *
Income (Rs)	30,000.00	26,000.00	30,000.00	0.85	0.65 ^NS^
PI	2.00	2.19	0.83	49.02	0.00 *
BOP	0.64	73.90	43.80	62.78	0.00 *
PPD(mm)	6.15	6.43	2.34	50.19	0.00 *
CAL(mm)	7.80	6.76	2.34	52.13	0.00 *
TC (mg/dL)	176.00	172.00	175.00	0.39	0.82 ^NS^
HDL (mg/dL)	32.00	53.00	52.00	29.23	0.83 ^NS^
LDL (mg/dL)	90.00	97.00	97.00	0.00	0.99 ^NS^
TG (mg/dL)	128.00	107.00	106.00	2.59	0.27 ^NS^
RBS (mg/dL)	142.00	103.00	102.00	30.23	0.00 *
BP(mmHg)	105.00	90.00	93.33	20.62	0.00 *

Level of significance—*p* < 0.05; *p*-value *—significant; *p*-value ^NS^—not significant. Footnotes: BMI—body mass index; PI—plaque index; BOP—bleeding on probing; PPD—probing pocket depth; CAL—clinical attachment level; TC—total count; HDL—high density lipoproteins; LDL—low density lipoproteins; TG—triglycerides; RBS—random blood sugar; BP—blood pressure; GP—generalized periodontitis; CAD—coronary artery disease; HP—healthy patients.

**Table 2 materials-14-04692-t002:** (**a**): Comparison of microRNA146a (subgingival plaque samples) among the three groups; (**b**): intergroup comparison of mean difference and test of significance of microRNA-146a (subgingival plaque samples) between the three groups.

(a)				
MicroRNA-146a Levels	Mean ± Std Dev	F-Value	*p*-Value	
(GP+CAD)a	2.55 ± 0.74			
GP	2.24 ± 0.54			
HP	1.21 ± 0.25	39.83	<0.001 *	
**(b)**				
**Variables**	**Groups**		**Mean Difference**	***p*-Value**
	(GP+CAD)a	GP significance—p < 0.05	0.31	0.123 ^NS^
MicroRNA-146a		HP	1.33	<0.001 *
	GP	HP	1.02	<0.001 *

Level of significance—*p* < 0.05; *p*-value *—significant; *p*-value ^NS^—not significant. Footnotes: GP—generalized periodontitis; CAD—coronary artery disease; HP—healthy patients.

**Table 3 materials-14-04692-t003:** (**a**): Comparison of microRNA126 (subgingival plaque samples) among the three groups; (**b**): intergroup comparison of mean difference and test of significance of microRNA-126 (subgingival plaque samples)between the three groups.

(a)					
MicroRNA 126 Levels	Mean ± Std dev		F-Value	*p*-Value	
(GP+CAD)a	1.36 ± 0.41		60.981	<0.001 *	
GP	1.84 ± 0.69				
HP	3.21 ± 0.70				
**(b)**					
**Variables**		**Groups**		**Mean Difference**	***p*-Value**
MicroRNA-126		(GP+CAD)a	GP	−0.48	0.19 ^NS^
			HP	−1.85	<0.001 *
		GP	HP	−1.37	<0.001 *

Level of significance—*p* < 0.05; *p*-value *—significant; *p*-value ^NS^—not significant. Footnotes: GP—generalized periodontitis; CAD—coronary artery disease; HP—healthy patients.

**Table 4 materials-14-04692-t004:** Pearson’s correlation between microRNA146a and microRNA126 between group Ia (subgingival plaque) and group Ib (coronary sample).

(GP+CAD)a (Subgingival Plaque Sample)	(GP+CAD)b	r-Value	*p*-Value
microRNA-146a	micro RNA-146a	−2.40	0.24 ^NS^
microRNA-126	micro RNA-126	0.12	0.57 ^NS^

Level of significance—*p* < 0.05; *p*-value ^NS^—not significant. Footnotes: GP—generalized periodontitis; CAD—coronary artery disease.

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
