# Peer review of "miRNA-146a and miRNA-126 as Potential Biomarkers in Patients with Coronary Artery Disease and Generalized Periodontitis"

_materials, 2021, doi:10.3390/ma14164692_

Round 1

Reviewer 1 Report

The work is good and focuses on patients with subgingival plaques. The author used microRNA to explore the relationship between CAD and generalized periodontitis. However, the author did not provide enough information about why they choose two microRNA (miRNA-146a and miRNA-126). The author should increase the introduction to let the readers understand this point. Did the author perform the whole gene survey of subgingival plaques to find the target microRNA? Upregulation of miRNA 146a and downregulation of micro-RNA 126 were found to be highly expressed in periodontitis patients associated with coronary artery disease as compared to other groups. The patients with only subgingival plaques without CAD also presented upregulation of miRNA 146a and downregulation of micro-RNA 126. Therefore, the mechanism of plaque is similar due to inflammation. The author may modify the conclusion. The author did not provide any information on other comorbidities (DM, HTN, etc) between the three groups. 

Author Response

Reviewer 1:

The work is good and focuses on patients with subgingival plaques. The author used microRNA to explore the relationship between CAD and generalized periodontitis. However did the author perform the whole gene survey of subgingival plaques to find the target microRNA? Upregulation of miRNA 146a and downregulation of micro-RNA 126 were found to be highly expressed in periodontitis patients associated with coronary artery disease as compared to other groups. The patients with only subgingival plaques without CAD also presented upregulation of miRNA 146a and downregulation of micro-RNA 126. Therefore, the mechanism of plaque is similar due to inflammation. The author may modify the conclusion. The author did not provide any information on other comorbidities (DM, HTN, etc) between the three groups.

Q 1: The author did not provide enough information about why they choose two microRNA (miRNA-146a and miRNA-126). The author should increase the introduction to let the readers understand this point.

Answers

Based on the reviewers suggestions the inclusion of miRNA-146a and miRNA-126 in the study has been elucidated and highlighted in the manuscript in the introduction section

Q 2

However did the author perform the whole gene survey of subgingival plaques to find the target microRNA?

Answers

The reviewer’s suggestions are well taken. The aim of the study was to quantify and compare the levels of micro-RNA 146a and micro- RNA 126 in subgingival as well as coronary plaque samples. In future, the whole gene survey of subgingival plaques to find the target microRNA will be taken for consideration for further research as per the suggestions.

Q 3

The patients with only subgingival plaques without CAD also presented upregulation of miRNA 146a and downregulation of micro-RNA 126. Therefore, the mechanism of plaque is similar due to inflammation. The author may modify the conclusion.

Answers

Based on the reviewer’s suggestions the conclusion has been modified.

Q 4

The author did not provide any information on other comorbidities (DM, HTN, etc) between the three groups.

Answers

The other comorbidities have been mentioned and highlighted in the inclusion and exclusion criteria of materials and methods section of the manuscript as per the reviewer suggestions.

Reviewer 2:
The article titled “miRNA-146a and miRNA-126 as potential biomarkers in patients with Generalised periodontitis and coronary artery disease”, by Mahendra et al., has focused on patients with Generalized periodontitis and coronary artery disease by collecting plaque samples and performing RNA isolation and quantifying the miRNA levels. Additionally, several parameters have been recorded from patients/volunteers, such as Age, height, weight, Income, Plaque index, BOP, PPD etc. Collecting this kind of data is really important to study the cause of diseases. As a result, the data presented is valuable. However, as reviewer I have several concerns regarding the originality of the data and its presentation.

 Major Comments:

 Q 1: A similar study by Yagnik et al. 2019, have been in the same region of India almost during the same time interval, August/2017-August/2018 or June/2018. The same study measures similar parameters from patients and focuses only on microRNA-146a and observed the similar results. The authors do not site or mention aforementioned study.

Answer

In our previous study done by Yagnik et al. 2019, only the subgingival plaque samples were investigated for identification of miRNA146. However, in the current study we have investigated both miRNA146 and mi RNA126 from subgingival and coronary plaque samples from the CAD patients. As per the reviewer valuable suggestions, the study done by Yagnik et al. 2019 has been cited in the present manuscript.

Q 2 The authors discuss some terms such as CAD, but in materials methods use CHDa and CHDb. What is the difference between CHD and CAD?

Answer

As per the reviewer valuable suggestions CHD (coronary heart disease) has been replaced by CAD (coronary artery disease).

Q 3: As the level of miRNA-146a is studied in the previous paper, miRNA-126 is novelty; however, the authors do not discuss how and why they choose two miRNAs as candidate. It would be interesting to screen many miRNAs instead of 2, or discuss clearly why 2 miRNAs were chosen.

Answer: the miRNAs 126 and 146 a were chosen based on their sensitivity and specificity in relation to periodontal inflammation and CAD which has been substantiated by citing previous studies in the manuscript. According to the reviewer’s suggestions the novelty of choosing the above biomarkers has been included in the introduction section of the manuscript. In our future studies, we intend to screen various RNAs to explore their specific role in these inflammatory conditions.

Q 4: There are some p-values and statistical analyses, but it is not described well which statistical analyses have been used or how.

Answer: As per the reviewer’s suggestion, the statistical analysis has been reframed and the tests used have been elaborated and highlighted in the manuscript.

 Minor Comments:

Q1: The manuscript is not well written and requires proof reading.

Answer: The manuscript has been modified and re-written based on the reviewer’s suggestion.

Q2: Line-137/8: reads “microRNA146a and microRNA126 were isolated from coronary plaque and subgingival plaque samples using miRNA isolation kit*.”, I think it is RNA isolation.

Answer: As per the reviewer’s suggestion, microRNA isolation kit has been changed to RNA isolation kit.

In summary, as the reviewer I see major concerns with novelty of the data, several missing information (e.g. statistical analyses) and based on my major concerns, I do not suggest this manuscript to be accepted for publication.

References:

Yagnik et al. 2019; https://onlinelibrary.wiley.com/doi/epdf/10.1111/jicd.12442

Reviewer 2 Report

The article titled “miRNA-146a and miRNA-126 as potential biomarkers in patients with Generalised periodontitis and coronary artery disease”, by Mahendra et al., has focused on patients with Generalized periodontitis and coronary artery disease by collecting plaque samples and performing RNA isolation and quantifying the miRNA levels. Additionally, several parameters have been recorded from patients/volunteers, such as Age, height, weight, Income, Plaque index, BOP, PPD etc.

Collecting this kind of data is really important to study the cause of diseases. As a result, the data presented is valuable.

However, as reviewer I have several concerns regarding the originality of the data and its presentation.

Major Comments:

  • A similar study by Yagnik et al. 2019, have been in the same region of India almost during the same time interval, August/2017-August/2018 or June/2018. The same study measures similar parameters from patients and focuses only on microRNA-146a and observed the similar results. The authors do not site or mention aforementioned study.
  • The authors discuss some terms such as CAD, but in materials methods use CHDa and CHDb. What is the difference between CHD and CAD?
  • As the level of miRNA-146a is studied in the previous paper, miRNA-126 is novelty; however the authors do not discuss how and why they choose two miRNAs as candidate. It would be interesting to screen many miRNAs instead of 2, or discuss clearly why 2 miRNAs were chosen.
  • There are some p-values and statistical analyses, but it is not described well which statistical analyses have been used or how.

Minor Comments:

  • The manuscript is not well written and requires proof reading.
  • Line-137/8: reads “microRNA146a and microRNA126 were isolated from coronary plaque and subgingival plaque samples using miRNA isolation kit*.”, I think it is RNA isolation.

In summary, as the reviewer I see major concerns with novelty of the data, several missing information (e.g. statistical analyses) and based on my major concerns, I do not suggest this manuscript to be accepted for publication.

References:

Yagnik et al. 2019; https://onlinelibrary.wiley.com/doi/epdf/10.1111/jicd.12442

Author Response

(The authors gave the same response as above.)

Round 2

Reviewer 1 Report

The manuscript improved after revision. Upregulation of miRNA-146a and downregulation of micro-RNA-126 in patients with coronary artery disease and periodontitis was noted when compared to healthy controls. The miRNA-146a and miRNA-126 are present and involved in the disease process of periodontitis and coronary artery disease. This target needs a future study to explore the associated mechanisms.

Author Response

Academic editor:

The Authors are on the right track to better understand the molecular mechanisms that make up the pathogenesis of periodontal disease and other dys-inflammatory diseases such as coronary artery disease. However, there are constraints that have to be considered.

The most important concerns the criteria for defining whether or not a subject is affected by periodontal disease. The chapter "criteria for periodontitis" is not clear. The threshold was defined by Tonetti & Claffy (2005) and Page & Eke (2007) and, after diagnosis, the stage by Papapanou (2018).

Q1: Why four sites?  Is your threshold arbitrary and in your case it works so? Are you considering correctly only proximal sites, I suppose?

Ans: Based on the reviewer’s suggestion, the accurate explanation of the measurement of BOP has been incorporated into the manuscript with 4 sites/tooth in the entire dentition. Reference has also been quoted for the same and highlighted in the materials and methods section. Gingival bleeding is one of the cardinal symptoms reflecting inflammation in the periodontal tissues. The bleeding on probing percentage is a valuable diagnostic test. (Lang 1996) Hence, was taken into consideration.

Q2: Why PPD ≤ 3mm in every site in patients considered without periodontal disease? Only one proximal site (or two in the same tooth) is not significant. Why these choices?

Ans: According to Fischman et al, PPD <3mmin 30% of tooth sites is considered as healthy. Hence, taking this criteria into consideration, the PPD was recorded. The modification has been made in the manuscript with the appropriate reference.

Was a first phase of cause-related therapy (aimed at resolving acute oral matters, any gingivitis, failures to respect the biological width, etc.) carried out before diagnosis?

Ans: No periodontal intervention was done prior to the patient selection for the study as we consider it to hold a bias in the study protocol. Once the parameters were recorded the patients were referred for the periodontal management after the study.

It is then necessary to clarify whether both a subgingival and a coronary plaque sample were harvested or only a coronary plaque sample was taken from the GP-CADb patient group and clearly explain the choice

Ans: GP+CAD is a group consisting of the patients from whom we had harvested both subgingival plaque samples (GP+CADa) and coronary plaque samples (GP+CADb). The subdivision is to identify the plaques. This was done to clearly compare, correlate and elucidate the expression of microRNA 146a and 126 in coronary plaque and subgingival samples, which may pave a way to establishing these plaque samples as a highly specific sample for the expression of these transcriptomes in CAD patients.

Reviewer 1:

The manuscript improved after revision. Upregulation of miRNA-146a and downregulation of micro-RNA-126 in patients with coronary artery disease and periodontitis was noted when compared to healthy controls. The miRNA-146a and miRNA-126 are present and involved in the disease process of periodontitis and coronary artery disease. This target needs a future study to explore the associated mechanisms.

Ans: The reviewer’s suggestion is well taken. Future studies will be undertaken to assess the more intricate and specific mechanistic link involved in the role of these transcriptomes in CAD and periodontal disease.
